# Effects of Deoxynivalenol-Contaminated Diets on Metabolic and Immunological Parameters in Broiler Chickens

**DOI:** 10.3390/ani11010147

**Published:** 2021-01-11

**Authors:** Insaf Riahi, Virginie Marquis, Anna Maria Pérez-Vendrell, Joaquim Brufau, Enric Esteve-Garcia, Antonio J. Ramos

**Affiliations:** 1Institute of Agrifood Research and Technology (IRTA Mas Bové), Animal Nutrition Department, 43120 Constanti, Spain; anna.perez@irta.cat (A.M.P.-V.); joaquim.brufau@irta.cat (J.B.); enric.esteve@irta.cat (E.E.-G.); 2Phileo by Lesaffre, 137 Rue Gabriel Péri, 59700 Marcq en Baroeul, France; v.marquis@phileo.lesaffre.com; 3Applied Mycology Unit, Food Technology Department, University of Lleida, UTPV-XaRTA, Agrotecnio, Av.Rovira Roure 191, 25198 Lleida, Spain; antonio.ramos@udl.cat

**Keywords:** deoxynivalenol, broiler chickens, deoxynivalenol-3-sulphate, blood hematology, immune response

## Abstract

**Simple Summary:**

Mycotoxin contamination in feed is a significant problem worldwide because these toxic metabolites can have serious adverse effects on animals, resulting great economic loss. Deoxynivalenol (DON) is the most commonly detected mycotoxin in cereals and therefore in poultry feed. This mycotoxin could adversely affect the health of birds. In this work, broiler chickens were exposed to two different levels of DON (5 and 15 mg/kg feed) for 42 days. The results revealed that broilers fed the high level (15 mg/kg) had DON-3-sulphate (a specific metabolite of DON) deposited in the liver and had reduced blood hematological parameters. The ingestion of the guidance level (5 mg DON/kg fee) affected the immune system parameters in broiler chickens. Both levels did not adversely affect the broiler’s welfare related-parameters.

**Abstract:**

The current study was conducted to examine the effects of deoxynivalenol (DON) at different levels (5 and 15 mg/kg feed) on the metabolism, immune response and welfare parameters of male broiler chickens (Ross 308) at 42 days old. Forty-five 1 day-old broiler chickens were randomly distributed into three different dietary treatments: (1) control, (2) DON-contaminated diet with 5 mg DON/kg of feed (guidance level), and (3) DON-contaminated diet with 15 mg DON/kg of feed. Five replicated cages with three birds each were used for each treatment in a randomized complete block design. The results showed that DON was detected in excreta of birds fed contaminated diets compared with controls. The metabolite DON-3 sulphate (DON-3S) was detected in plasma and excreta in both treated groups, as well as in the liver (but only at 15 mg/kg feed). The increase in the level of DON decreased the hemoglobin concentration (*p* < 0.001), whereas the erythrocyte counts were only decreased at 15 mg DON/kg feed. No effect of DON on the responses to common vaccines was observed. In plasma, interleukin 8 levels in both contaminated groups were significantly higher than in the control group. The expression of interleukin 6, interleukin 1β and interferon-γ increased in jejunum tissues of broilers fed 5 mg/kg of DON compared with controls. The stress index (heterophil to lymphocyte ratio) was not affected by DON-contaminated diets compared with controls. The plasma corticosterone level was significantly lower in both DON groups compared with controls. In conclusion, DON-3S could be used as a specific biomarker of DON in different biological matrices, while the immune response in broiler chickens is stimulated by the presence of DON at the guidance level, but no adverse effect was observed on physiological stress parameters.

## 1. Introduction

Deoxynivalenol (DON) is a secondary toxic metabolite mainly produced by *Fusarium* species that belongs to the trichothecenes family. DON frequently occurs in cereals, including wheat, maize, barley, rye and oats [1]. A 10-year survey from 2008 to 2017 of the global mycotoxin occurrence in feed revealed that DON was the most prevalent of mycotoxins and was detected in 64% of 74,821 samples collected from 100 countries [2].

Therefore, DON is considered to be the most frequently found mycotoxin in poultry feed, as chicken diets consist of high levels of cereals. The guidance level of DON in poultry feed is 5 mg/kg [3]. In term of productive parameters, it has been reported in some studies that poultry could tolerate up 15 mg/kg feed [4,5]. This tolerance could be related to the metabolism of DON in this specie [6]. The metabolism of DON or other mycotoxins is defined as the conversion of the native toxin (DON) to various degradation metabolites in the organism and in the digestive tract by microbes [7]. The analysis of the metabolites in the plasma of broiler chickens after a single intravenous injection or oral bolus of the synthetic or labeled DON at the guidance level, revealed that the main metabolite is DON-3-sulphate (DON-3S) of all metabolites [6,8]. However, the determination of DON-3S in plasma, liver and excreta of broiler chickens fed chronic DON at the guidance level has not been evaluated to date. Knowledge on the metabolites of mycotoxins is essential to test the efficacy of detoxifying agents in vivo afterwards [9].

On the other hand, the immune system is a target of DON mycotoxicosis [10]. As with other trichothecenes, DON can induce either immunostimulation or immunosuppression, depending on the dose and the duration of exposure [11]. A Low to moderate concentration induces the up-regulation of cytokines, whereas a high concentration induces the apoptosis of immune cells [12]. However, results are not conclusive regarding the effect of DON on parameters related to the poultry immune system. Furthermore, DON-induced physiological stress has been observed in chickens, but few studies have evaluated the related stress indicators [13,14].

The purposes of this study were to investigate the metabolism of DON in broiler chickens after chronic feeding and to better understand its effects on chicken immune response at both 5 mg/kg and 15 mg/kg levels. We also tested the hypothesis that DON can affect the physiological stress parameters of birds.

## 2. Materials and Methods

### 2.1. Ethical Approval

All animal care procedures were approved by the Ethical Committee for Animal Experimentation of IRTA, in accordance with current regulations on the use and handling of experimental animals (Decree 214/97, Generalitat de Catalunya, Catalonia, Spain).

### 2.2. Birds, Diets, and Experimental Design

Birds, management, the production of DON and contaminated diets, and experimental design for this current study have already been detailed in a recently published paper [15]. Forty-five 1-day-old male chicks (Ross 308) were randomly allotted to 15 battery cages (0.62 m × 0.62 m × 0.41 m). Birds were vaccinated against infectious bronchitis virus (IBV) at the hatchery and against Newcastle disease virus (NDV) at the farm at start of the trial. At the farm, the lighting program was 24 h of light per day for the first two days, 18 h of light per day until 7 days, and 14 h of light per day afterwards. The birds were maintained at 34 °C for the first two days and the temperature was gradually decreased by 3 °C per week until reaching 21 °C, and then was maintained. Chickens were fed starter diets from 1 to 21 days and grower diets from 22 to 42 days based on corn, soybean meal, soy oil, and a premix with vitamins, minerals, and amino acids; diets were formulated according the nutrient requirements for Ross 308 strain broilers (Table 1). Feed was provided in mash form *ad libitum*, in a metal feeder connected to each battery cage. Unlimited access to water was provided from individual nipple drinkers. Three dietary treatments with five cages per treatment and three birds per cage were used. The treatment 1 received a non-contaminated diet, and treatments 2 and 3 received DON contaminated diets at 5 mg/kg feed or 15 mg/kg feed, respectively for 42 d. DON used in this trial was produced by inoculating wheat with *Fusarium graminearum* strain I159 as reported by Metayer et al. [16] (ENVT, Toulouse, France).

### 2.3. Analysis of Mycotoxins in Experimental Feeds

The presence of DON and other mycotoxins in the feeds used in the assay was evaluated. Regarding DON analysis, 5 g of ground feed from each diet was mixed with 40 mL of distilled water and stirred for 10 min at 600 rpm. Thereafter, the mixture was filtered through Whatman paper no4, and 2 mL as passed through a DONPREP^®^ immunoaffinity column (r-Biopharm, Rhone LTD, Glasgow, UK) which was cleaned up with 5 mL of MiliQ water. To eluate DON, 3 mL of HPLC-grade methanol was passed through the column and evaporated to dryness under a gentle stream of nitrogen. Dry extract was reconstituted in 1 mL of HPLC-grade mobile phase and 100 μL was analyzed by HPLC using a Waters (Milford, MA, USA) Module Alliance 2695^®^, coupled to a UV/Visible dual λ absorbance Detector Waters 2487^®^. A Waters Spherisorb^®^ 5 μm ODS2, 4.6 × 250 mm column was used. Absorption wavelength was set at 220 nm. The mobile phase was methanol:acetonitrile:water (4:4:92, *v/v/v*) and was set at a flow rate of 1.2 mL min^−1^, and the column temperature was set at 40 °C. The limit of detection (50 µg/kg) was considered to be three times the signal of the blank.

With regard to Aflatoxin B_1_ (AFB1) analysis, 5 g of ground feed sample from each diet was mixed with 15 mL of 60% methanol and stirred for 10 min at 600 rpm. Thereafter, the mixture was filtered through Whatman paper nº 4, and 2 mL of filtrate were added to 14 mL of phosphate buffered saline (PBS) and mixed well. The whole solution was passed through an Easy-extract^®^ Aflatoxin immunoaffinity column (r-Biopharm), which was cleaned up with 20 mL of PBS. To eluate AFB1, 1.5 mL HPLC-grade methanol and 1.5 mL MiliQ water were sequentially passed through the column and 100 μL of the joint eluates was analyzed by HPLC coupled with a fluorescence detector (FLD). The chromatographic equipment and column were the same as those used for DON analysis, but coupled to a Multi λ Fluorescence Detector Waters 2475^®^. The excitation wavelength was set at 365 nm and the emission wavelength was set at 465 nm. The derivatization of AFB1 was obtained using a post-column photochemical derivatization device (UVE™ Derivatizer LC Tech). The mobile phase consisted of a solution of water:methanol:acetonitrile (70:17:17) and was set at a flow rate of 1.2 mL min^−1^, and the column temperature was set at 40 °C. The limit of detection (0.3 µg/kg) was considered to be three times the signal of the blank.

The detection of zearalenone (ZEN), total fumonisins (FBs) and ochratoxin A (OTA) in feed samples was carried out using the Ridascreen^®^ Zearalenon, Ridascreen^®^ Fumonisin and Ridascreen^®^ Ochratoxin A enzyme-linked-immunosorbent assay (ELISA) kits (R-Biopharm), following the manufacturer’s instructions, with detection limits of 1.75, 25, and 2.5 µg/kg for ZEN, FBs and OTA, respectively. All mycotoxin analyses were carried out by the Applied Mycology Unit of the Food Technology Department of the University of Lleida (Spain).

### 2.4. Sampling and Analysis

At 42 days of age, blood samples (3 mL/bird, 3 birds/pen) were collected by cardiac puncture in non-heparinized tubes for the hematological and serological analysis. Blood samples (3 mL/bird, 3 birds/pen) were collected also into heparinized tubes for interleukin 8 (IL-8), corticosterone, DON and DON-3S determination (Table 2). Blood serum of each bird was separated by centrifugation at 4500× *g* for 10 min. Plasma was separated by centrifugation at 1000× *g* for 15 min for IL-8 and corticosterone determination and at 2851× g for 10 min for DON and DON-3S determination. Samples were stored at −20 °C until further analysis. Mortality rates were 13%, 20% and 13% for the control treatment, and treatments 2 and 3, respectively. Twelve birds in each treatment were idividually weighed and humanely euthanized according to IRTA ethics instructions at 42 days. Immediately, the entire intestine was carefully removed and the distal part of the jejunum (5 cm taken from Meckel’s diverticulum) was collected for each bird, rinsed in PBS, and subsequently stored in RNAlater (Vidra Foc, Barcelona, Spain) for 24 h at ambient temperature. Then, samples were stored at −80 °C without RNAlater until quantitative real-time PCR (qRT- PCR) analysis. Liver samples were excised, weighed, and stored at −20 °C until lyophilization. Fresh excreta samples from each cage were collected daily (day 1 to day 42) and stored at −20° C until lyophilization. One cage sample was a pool of excreta of three birds. Lyophilized liver and excreta samples were stored at darkness at ambient temperature until further analysis.

### 2.5. DON and DON-3S Determination in Different Biological Matrices (Plasma, Liver, and Excreta)

#### 2.5.1. Chemicals, Products and Reagents

The DON analytical standard was supplied by Sigma-Aldrich Chemie GmbH (Steinheim, Germany). The DON standard was dissolved in acetonitrile (ACN) as stock solution (1 mg/mL), and then diluted with HPLC-grade ACN to obtain an individual working standard solution of 1 µg/mL. The stable internal standard (IS) isotope (^13^C_15_-DON) was obtained from Romer labs (Bioser, Barcelona, Spain) as 1.2 mL of a solution of 25 µg/mL in ACN. An individual working standard solution of 5 µg/mL was prepared by diluting the above stock solution with HPLC-grade ACN and was stored at −15 °C. Acetic acid (LC-MS gradient grade) was purchased from Montplet & Esteban SA (Barcelona, Spain). Sodium sulfate, sodium acetate and hexane were from PanReac Quimica SLU (Barcelona, Spain). Ammonium formate was from Sigma-Aldrich Chemie GmbH (Steinheim, Germany). Methanol (HPLC gradient grade, MeOH) was purchased from Honyewell (Seelze, Germany).

#### 2.5.2. In Plasma

Plasma extract preparation was carried out according to the method of Broekaert et al. [17]. Briefly, 5 μL of ^13^C_15_-DON IS solution (at 1 µg/mL and 750 μL of ACN were added to 250 μL of chicken plasma. ACN was added to precipitate plasma proteins. The samples were vortexed approximately for 1 min. Afterwards, the samples were centrifuged at 8517× *g* for 10 min. The supernatant (1 mL) was transferred to a new tube, evaporated to dryness under nitrogen flow over a heating block and reconstituted in 1 mL of ammonium formate 5 mM/MeOH (50:50, *v*/*v*) solution, and 10 μL was injected for HPLC-MS/MS analysis.

#### 2.5.3. In Liver and Excreta

Samples of lyophilized liver or excreta were weighed (1 g) in 50 mL centrifugation tubes. In total, 5 μL of 1 µg/mL of IS working solution (^13^C_15_-DON) and 10 mL of ACN: water: acetic acid (79:20:1, *v*/*v*/*v*) were added. Samples were vortex mixed for 2 min. Four grams of sodium sulphate and 1.5 g of sodium acetate were added and then each tube was vortex mixed for 5 min and then with an orbital shaker for 20 min (IKA™ KS 260, Fisher Scientific, Madrid, Spain). After extraction, a centrifugation was made at 2716× *g* for 10 min, and supernatants were transferred to another tube and 5 mL of hexane was added and vortexed. After the separation of two phases, the hexane phase was removed. A 5 mL aliquot of the extract was evaporated to dryness under nitrogen flow over a heating block and reconstituted in 0.5 mL of water/ammonium formate (5 mM):MeOH (50:50, *v*/*v*) and then filtrated through nylon syringe filters (0.20 µm) from Agilent (Santa Clara, CA, USA) and injected directly in HPLC-MS/MS.

#### 2.5.4. LC-MS/MS Analysis

The HPLC-MS/MS analysis was carried out with a Transcend 600 LC (Thermo Scientific TranscendTM, Thermo Fisher Scientific, San Jose, CA, USA) coupled to an Orbitrap (ExactiveTM, Thermo Fisher Scientific, Bremen, Germany) with an electrospray ionization (ESI) source (HESI-II, Thermo Fisher Scientific, San Jose, CA, USA). Chromatographic separation was achieved using a Zorbax Plus C18 (1.8 μm × 2.1 × 100 mm) column from Agilent (San Jose, CA, USA). Gradient elution was established with a mobile phase consisting of 5 mM ammonium formate in water (eluent A) and methanol (eluent B) at a flow rate of 0.2 mL/min. The gradient elution started at 95% B at 1 min and was decreased to 0% B at 8 to 12 min afterwards; it increased to 95% at 12.5 min, which was maintained up to 14 min. The column temperature was set at 25 °C and the injection volume was 10 µL. MS analyses were performed using a selected reaction monitoring (SRM) mode with positive and negative electrospray ionization (ESI±). The settings on the spectrometer were as follows: compounds were ionized by electrospray ionization in the positive and negative mode, measured first in full scan, and then in targeted MS/MS mode at a collision energy of 30 eV (both in the range from m/z 50–500). ESI parameters were as follows: spray voltage, 4 kV; sheath gas (N2, >95%), 35 (adimensional); auxiliary gas (N2, >95%), 10 (adimensional); skimmer voltage, 18 V; capillary voltage, 35 V; tube lens voltage, 95 V; heater temperature, 305 °C; capillary temperature, 300 °C. The capillary and nozzle voltage were 4000 V and 95 V (−95 V in ESI-), respectively. Finally, the data were processed using XcaliburTM version 3.0 (Quanbrowser and Qualbrowser) and Mass FrontierTM 7.0.

### 2.6. Blood Hematology

Hemoglobin (HGB, g/dL) and erythrocytes (Red Blood Cells, RBC/µL) were measured using a CELL-DYN 3700 hematology analyzer (Abbott, Chicago, IL, USA). A blood sample was collected in microcentrifuge (Haematokrit 200, Helltich Zentrifugen, Tuttlingen, Germany) capillary tubes for hematocrit (HCT, %) determination, which was performed in a Neubauer chamber (Brand, Germany). The mean corpuscular volume (MCV, fL) and mean corpuscular hemoglobin (MCH, pg) were determined by the hematologic analyzer CELL-DYN 3700 (Abbott, Chicago, IL, USA), and mean corpuscular hemoglobin concentrations (MCHC, g/dL) were calculated as MCHC = hemoglobin/hematocrit. Leukocytes per µL and the differential leukocyte count (heterophils, lymphocytes, monocytes, eosinophils and basophils, %) were also measured using a hematologic analyzer CELL-DYN 3700 (Abbott, Chicago, IL, USA).

### 2.7. Response to Common Vaccines (NDV and IBV)

Antibody titers against NDV or IBV were determined by the hemagglutination inhibition (HI) test using standard protocols by the World Organisation for animal health (OIE). Serial twofold serum dilutions were made in PBS and 0.025 mL was added to the wells. Four hemagglutination (HA) units of the test antigen were added to each dilution and incubated at room temperature for 30 min. An equal volume of 1% chicken red blood cells (RBCs) in PBS was then added to the wells until agglutination occurred in the negative control sample. All plates included NDV or IBV-negative and NDV or IBV-positive control sera. The highest dilution of serum displaying the inhibition of agglutination was designated as the reciprocal log2 HI titer for that serum sample. Moreover, IBV was determined in serum using an ELISA test kit (Idexx^®^, Westbrook, ME, USA) according to the protocols specified by the supplier. Briefly, 96-well plates were coated with viral antigen; after the incubation of the test sample (100 µL) in the coated well, an IBV-specific antibody formed a complex with the coated viral antigens. After washing away unbound material from the wells, a conjugate (100 µL) was added which bound to any attached chicken antibody in the wells. Unbound conjugate was washed away and 100 µL of substrate solution (TMB) was added to the wells and incubated for 15 min at ambient temperature. Subsequent color development was directly related to the amount of IBV antibody present in the test sample. The color development was stopped with a stop solution (100 µL) and the absorbance values were measured and recorded at 650 nm. Then, the amount of antibody to IBV present in the test sample was calculated.

### 2.8. Plasma IL-8 Determination

IL-8 was determined in plasma using a commercially available ELISA kit (chicken IL-8 ELISA kit) according to the manufacturer’s instructions (MyBioSource, San Diego, CA, USA). Briefly, 96-well plates were pre-coated with anti-IL-8 antibody, and the biotin conjugated anti-IL-8 antibody was used for the detection of antibodies. The standards, test samples, and biotin-conjugated detection antibody (50 µL each) were subsequently added to the wells and washed with wash buffer. Streptavidin–horseradish peroxidase enzyme (HRP) (50 µL) was added and unbound conjugates were washed away with wash buffer. TMB (3, 3′, 5, 5′ tetramethylbenzidine) substrates were used to visualize the HRP enzymatic reaction. TMB was catalyzed by HRP to produce a blue-color product that changed to yellow after adding an acidic stop solution. As the density of yellow was proportional to the IL-8 amount of the sample captured on the plate, the optical density absorbance at 450 nm in a microplate reader (Anthos, Labtec instruments, Salzburg, Austria) was read, and the concentration of IL-8 was calculated.

### 2.9. Gene Expression by Quantitative Real-Time PCR (qRT-PCR)

Total RNA from the tissue samples of the distal jejunum (20 mg) was extracted using an RNeasy Mini Kit (Qiagen, Hilden, Germany) according to the manufacturer’s instructions. RNA was eluted into 50 μL of Rnase free water and stored at −80 °C. The yield of RNA was determined by spectrophotometry (BioPhotometer, Eppendorf, Hamburg, Germany). Cytokine gene expression (IL-6, IL-1β, IFN-γ and IL-10) were evaluated as previously described by Reid et al. [18]. The mRNA quantification of cytokines was determined by qRT-PCR using QuantiTect™ SYBR^®^ Green one-step RT-PCR Kit (QIAGEN, Hilden, Germany). The PCR amplification was performed using 7500-Fast Real-time PCR (Applied Biosystems, CA, USA) [18].The threshold cycle values (Ct) were t normalized to the reference gene (glyceraldehyde-3-phosphate dehydrogenase (GAPDH)). The average ∆Ct of the control samples was used to calculate the target gene expression according to the 2^−∆∆Ct^ method [19]. This relative quantification related the PCR signal of the target transcript gene in a treatment group to the average signal of untreated control. Duplicate samples were used.

### 2.10. Physiological Stress Related-Parametrs

#### 2.10.1. Stress Index (Heterophil to Lymphocyte Ratio)

The heterophil to lymphocyte ratio (H/L), considered as an indicator of stress, was calculated by dividing the number of heterophils by the number of lymphocytes [13,20].

#### 2.10.2. Plasma Corticosterone Determination

The plasma level of corticosterone was measured with a commercially available ELISA kit (chicken corticosterone ELISA kit) according to the manufacturer’s instructions (Cusabio, Houston, TX, USA). Briefly, the microtiter plate provided in this kit was pre-coated with an antigen. Standards, samples, antibody specific for corticosterone (CORT) (50 µL each), and HRP-conjugate (100 µL) were added to the appropriate microtiter plate wells. The competitive inhibition reaction was launched between pre-coated CORT and CORT in samples. Then, 100 µL of substrate solution (TMB) was added to the wells, incubated for 15 min at 25 °C, and protected from light. The color development was stopped with an acid solution (50 µL) and the intensity of the developed yellow color was measured. A microplate reader (SkanIt, Thermo Fisher Scientific, Madrid, Spain) capable of measuring absorbance at 450 nm was used.

### 2.11. Statistical Analysis

Statistical analysis was performed by SAS software (SAS 9.4, SAS Institute, Cary, NC, USA). After the determination of normality and variance homogeneity, data were evaluated as a completely randomized design by a one-way analysis of variance (ANOVA) using the General Linear Model Procedure to test the effect of different treatments. Response to common vaccine parameters were logarithmically transferred to maintain the homogeneity of variance. Each cage was considered to be an experimental unit. Results were shown as means ± standard error of the means (SEM). Orthogonal polynomials were used to determine linear and quadratic dose responses. To test the normal distribution of data, a Kol–Mogrov–Smirnov test was used. The significance level was set at *p* ≤ 0.05. A trend was defined as *p*-value between 0.05 and 0.10 (0.05 < *p* ≤ 0.10).

## 3. Results

### 3.1. Dietary Mycotoxin Concentrations

The DON levels found in control diets were 65 and 73 µg/kg in the starter and grower diets respectively. The concentrations of DON in contaminated feeds were close to 5 and 15 mg/kg, as expected. Diets also included lesser amounts of ZEN, FBs and OTA. AFB1 was not detected in all experimental diets (Table 3).

### 3.2. DON and DON-3S Determination in Plasma, Liver, and Excreta

After a chronic DON feeding of broilers at a low (5 mg/kg) or high level (15 mg/kg) for 42 days, DON and DON-3S were analyzed in plasma, liver and excreta. Results showed that, in the non-contaminated control treatment, DON was below the limit of quantification (LOQ) (5 ng/mL) and DON-3S could not be identified. Similarly, DON was below the LOQ (5 ng/mL) in plasma and liver in broilers fed a contaminated diet at both levels but was detected only in excreta. DON-3S was detected in plasma and excreta at both levels and was significantly lower in the DON low dosage group. Interestingly, DON-3S was also detected in liver but only at the highest level assayed (Table 4).

### 3.3. Hematological Indices

The results of blood hematology paramters are reported in (Table 5). DON contaminated feed decreased the HGB level in a dose dependent mannner (*p* = 0.0002). Furthermore, the presence of DON at 5 mg/kg did not affect RBC, MCV, and MCHC (*p* > 0.05). However, 15 mg/kg DON in broiler diets significantly reduced RBC and MCHC and increased MCV blood level (*p* < 0.05). Leukogram data were not affected by the different dietary treatments (*p* > 0.05).

### 3.4. Response to Common Vaccines (NDV and IBV)

No significant effects of dieatray treatments were observed on titers against NDV and IBV in broilers at 42 days (*p* > 0.05) (Table 6)

### 3.5. Plasma IL-8 Production and Realtive mRNA Expression of Immune Genes

In plasma, IL-8 was significantly up-regulated in all broiler chickens receiving DON (5 and 15 mg/kg) compared to the control group (*p* = 0.001) (Figure 1). DON feeding at 5 mg/kg significantly stimulated the mRNA relative expression of IL-6, IFN-^γ^, and IL-1β in the jejunal tissues of broiler chickens (*p* < 0.05). However, the mRNA expression of these genes was comparable to the control group when the diet was contaminated with 15 mg/kg feed (Figure 2).

### 3.6. Physiological Stress Parameters

The effects of experimental treatments on welfare-related indicators are presented in Table 7. No significant differences among the diet groups were detected for the stress index (H/L ratio). However, birds fed DON at both levels (5 and 15 mg/kg) showed lower plasma corticosterone level than birds fed a control diet (*p* = 0.03).

## 4. Discussion

In practice, it is almost impossible to find feed that does not have a basal DON contamination, given the frequent contamination of cereals with this mycotoxin. Thus, the dietary concentration of DON was 65 and 73 µg/kg in the starter and grower control feed, respectively. This level was too low to cause adverse effects in broiler chickens [3]. Dietary ZEN concentrations ranged between below the LOQ and 259 µg/kg and increased as DON increased in the diet. DON and ZEN can be found alone or together [21]. According to literature reports, the content of ZEN in the present study was not enough to produce synergism with the amounts of DON that negatively affect the health of broiler chickens [22]. Besides, FBs and OTA levels were below the guidance value established for poultry feed (20 and 0.1 mg/kg for FBs and OTA, respectively) [3]. Furthermore, the levels of AFB1 were below the limit of detection in all feeds. Therefore, the effects observed in the evaluated indicators in the current study were not attributable to AFB1, ZEN, FBs, and OTA, and thus, only DON was responsible for these effects.

Although other authors have stated that the concentrations of DON chosen for this study should not have major effects on performance [4,5], our results revealed that 15 mg/kg reduced body weight gain and altered the feed conversion ratio of broiler chickens at 42 days [15].

Regarding the pathway of metabolization, DON was not detected in the plasma of birds even at 15 mg/kg feed. This result could be explained by the low absorption of this mycotoxin into plasma. The absolute oral bioavailability at the guidance value in chickens fed orally is poor, amounting to only 19.3% [23]. Similarly, DON was not detected in the liver. This result could be related to its rapid metabolism and excretion. Other researchers explain the lack of quantification of DON in plasma and liver by the protective effect of the hepatic/renal first pass effect. In fact, poultry are characterized by the protective hepatic/renal first pass effect. This effect is controlled by gut and liver enzymes, which induce the oxidation, reduction or hydrolysis (phase I reactions), and/or conjugation (phase II reactions) of toxins [24]. However, our results revealed that DON could be quantified in excreta of birds fed DON-contaminated diets. The detection of DON in excreta may be attributed to the rapid clearance of this toxin into excreta. The mentioned findings were previously also described when broilers were fed DON at 5 mg/kg by Awad et al. [25].

Toxicokinetic studies performed in chickens (after single intravenous or oral bolus injection of labeled or synthetic DON) have demonstrated that DON-3S is the most abundant metabolite in plasma and excreta [6,8,26,27]. In addition, only one chronic study evaluated DON-3S in excreta of broiler chickens after chronic feeding of DON at a lower concentration than the maximum recommended (1.7 mg/kg) [28]. In the current study, DON-3S was detected in plasma and excreta of broilers exposed to recommended levels and at 15 mg/kg, suggesting the extensive metabolization of DON to DON-3S in broiler chickens. Interestingly, DON-3S was detected in liver, but only in broilers exposed to the highest concentration tested. This result could be associated to the bird’s cholesterol metabolism, as the blood cholesterol level was significantly affected in birds fed 15 mg DON/kg [15]. This result could correlate with the adverse effect of 15 mg DON/kg on performance found in a recent research work from our group [15], and may confirm that, if feed contamination is lower (5 mg/kg), birds continue to have sufficient capacity for excretion and a limited deposition into the liver. In conclusion, DON-3S is a suitable biomarker for DON exposure in broilers; this biotransformation could be considered as a detoxification pathway [29] and may explain the low susceptibility of broilers to DON at guidance levels [26].

It has been suggested that the impairment of the immune system is the most important outcome of DON toxicity [10]. Consequently, it was expected that DON might affect blood hematological parameters, response to common vaccines (NDV and IBV), and cytokine production. In fact, the results of the current study showed that DON at higher levels (15 mg/kg) slightly more prominently affected hematological indices than 5 mg/kg feed. A significant dose-dependent decrease of hemoglobin concentration and a significant effect of a dietary dose of 15 mg DON/kg feed on RBC values, MCV, and MCHC were observed. The loss of HGB concentration and RBC count induced by DON mycotoxicosis could be a marker of bone marrow malfunction [30]. In fact, bone marrow is an immune organ that is susceptible to DON mycotoxin because cells rapidly divide in this organ. On the other hand, the values of hematological parameters observed were still within the range of reference values, and no anemia was induced [31]. The feeding of chickens with contaminated diets containing 9 or 18 mg of DON/kg of feed significantly decreased the hemoglobin concentration and the RBC count [32,33]. Moreover, it has been reported that DON could affect the humoral immune response by reducing the antibody titers against NDV and IBV [13,34]. However, in the current research, neither 5 nor 15 mg/kg of DON in broilers feed affected the vaccinal immune response after regular vaccination with NDV and IBV in broilers (*p* > 0.05). Yegani et al. [35] observed no effect of feeding broiler breeder hens with grains naturally contaminated with *Fusarium* mycotoxins based on DON (12.6 mg/kg) for 12 weeks on antibody titers against NDV. This result was also in agreement with those of Harvey et al. [33], who reported no effect on NDV antibody titers in White Leghorn chicks fed 18 mg of DON/kg of feed for 9 weeks. Regarding the antibody titers against IBV, Swamy et al. [22] found no effect on IBV titers with different concentrations of DON (4.7 and 8.3 mg/kg of feed) in broiler chickens exposed for 21 days and 42 days. Similar findings were reported by Yegani et al. [35], after feeding broiler breeder hens a concentration of 12.6 mg DON/kg for 28 and 56 days. Furthermore, no effects were found with a concentration of 12.2 mg DON/kg on antibodies against IBV titers in broilers chickens after 14 and 28 days [36]. The failure to observe significant results in response to common vaccines and the variability between literature reports suggests that those parameters could not be a relevant biomarker for DON toxicity in poultry.

IL-8 is a proinflammatory cytokine involved in pathogen defense and immune regulation, and it is considered as an early biomarker of the inflammation process [37]. In this study, DON presence in chicken feed at 5 and 15 mg/kg resulted in an increase of IL-8 production in plasma, suggesting that DON could have an effect on the innate immune response and inflammation process. The up-regulation of IL-8 has been shown in in vitro *and* in vivo studies in humans, rodents and farm animals [38,39]. To our knowledge, the up-regulation observed of the plasma IL-8 of broiler chickens after DON exposure is new information. The duration of exposure of DON is an important variable, as with a shorter duration (35 days), no significant differences between control and contaminated feeding groups were observed in a previous feeding trial with broiler chickens [40].

In the actual study, the effects of DON on the gene expression of cytokines in jejunum tissues of broilers were evaluated as markers of the intestinal immune system. The addition of 5 mg DON/kg feed significantly up-regulated the mRNA expression of the proinflammatory cytokines IL-6, IFN-γ, and IL-1β, suggesting that DON at the guidance level is immunostimulatory in broilers aged 42 days. These results are in agreement with previous results indicating that DON exposure from 2 to 5 mg/kg up-regulated the proinflammatory cytokines in broiler chickens, such as IL-6 in jejunum, IL6 and IL-1B in spleen, and IFN-γ in cecal tonsils [41,42,43]. The up-regulation of the mRNA gene expression of cytokines is due to the ability of DON, as a protein-synthesis inhibitor, to impair the synthesis of high turnover proteins and, as a result, to induce a transient expression of specific mRNAs [44]. In a similar manner, it has been suggested that this induction of immune-related genes by DON mycotoxin is due to the increasing of the binding activity of transcription factors in leukocytes such as the nuclear factor κB (NF-κB) at the transcription level, and to the increasing stability of the mRNA at the post-transcription level [11,42]. However, mRNA expression of IL-6, IFN-γ, and IL-1β was comparable to control group when broilers fed 15 mg/kg, suggesting that this dose did not induce immune-suppression in broilers aged 42 days. The mRNA level of the anti-inflammatory IL-10 was not statistically affected, suggesting that DON did not affect anti-inflammatory cytokines, as previously reported in some studies [42,43]. IL-10 is involved in the cross-regulation of IFN-γ gene expression [45]. The up-regulation of IFN-γ gene expression, therefore, may not be necessary to correlate the changes in IL-10 gene expression [42].

The physiological stress includes the elevation of the stress index (H/L ratio), defined as a result of the impairment of the number of circulating heterophils and lymphocytes, and an elevation of the circulating levels of corticosterone [46], although the effect of DON on the H/L ratio and plasma corticosterone level in poultry was not extensively documented [14,47]. As no significant effect was observed on blood heterophils and lymphocytes, therefore, dietary DON did not affect the stress index H/L ratio. Similarly, Danicke et al. [34] did not find significant differences in the H/L ratio of broilers fed a contaminated diet with a concentration of 14 mg/kg of DON per kg.

Corticosterone is the primary glucocorticoid secreted by the adrenal glands in birds and is involved in immune reactions and stress responses [48]. Corticosterone was selected as a physiological stress marker as the effects of feeding a contaminated diet containing DON was previously shown to increase this glucocorticoid in the plasma of broiler chickens [14,47]. The results of this study showed that the plasma levels of corticosterone decreased in broilers fed 5 and 15 mg/kg of DON compared to the controls. The differences between these results may be attributed to the very marked circadian cycle of the plasma corticosterone measurement. In fact, the difference of sampling time between treatments could affect the circadian cycle and thus could mask the real effect on the plasma corticosterone level. Furthermore, blood sampling is stressful to the animal and can mask the effect of the stressor under study.

## 5. Conclusions

In conclusion, DON and DON-3S in excreta are suitable metabolites of DON exposure in broilers fed the guidance level (5 mg/kg). Moreover, DON at this level induced proinflammatory cytokine production, suggesting that DON is immunostimulatory in broiler chickens at 5 mg/kg for 42 days. The effect of this high dose (15 mg/kg) was observed on the deposition of DON-3S in the liver and on the reduction in hematological parameters, suggesting that DON affects the health status of birds. Further studies are required to directly elucidate DON and DON-3S in excreta as relevant end-points for the efficacy testing of detoxifiers, and further investigations will need to pay closer attention to the most important immune system indicators that could be affected by DON mycotoxicosis in poultry.

## Figures and Tables

**Figure 1 animals-11-00147-f001:**
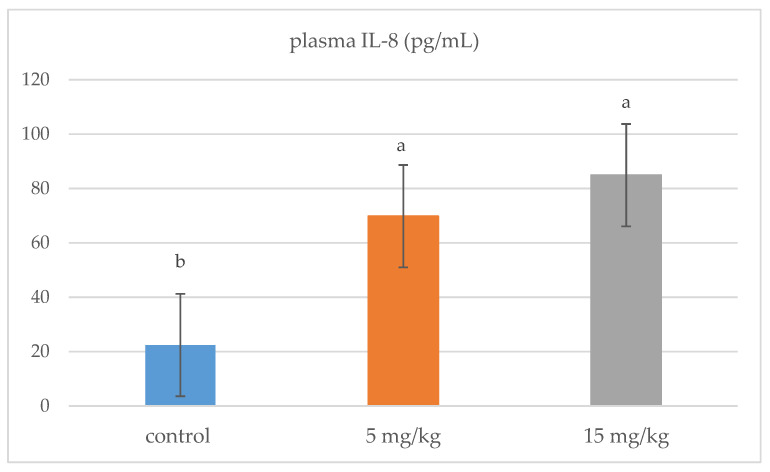
Effects of DON-contaminated feed (5 and 15 mg/kg) on plasma IL-8 levels in broiler chickens. Bars show the means and the standard error of mean (SEM) (*n* = 5); ^a,b,^: values with different superscripts for each cytokine differ (*p* ≤ 0.05).

**Figure 2 animals-11-00147-f002:**
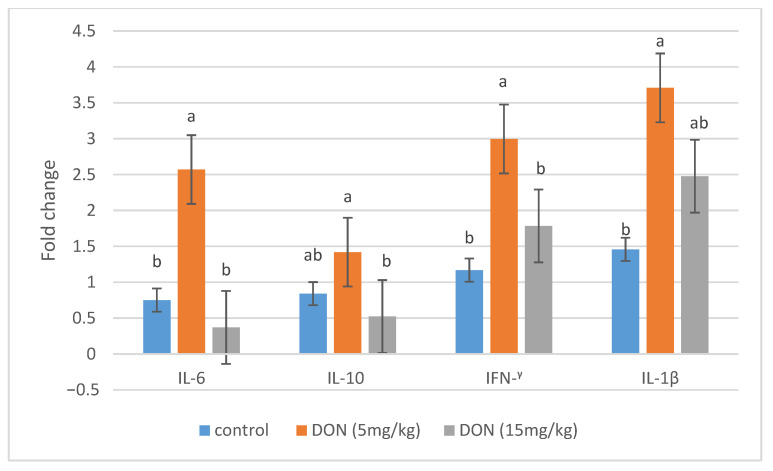
Effects of DON-contaminated feed (5 and 15 mg/kg) on the relative mRNA expression of immune genes (IL-6, IL-10, IFN-^γ^, and IL-1β) in the jejunal tissues of broiler chickens determined by qRT-PCR. Bars show the means and the standard error of means (SEM) (*n* = 5); ^a,b^: values with different superscripts for each cytokine differ (*p* ≤ 0.05).

**Table 1 animals-11-00147-t001:** Formulation and proximate analysis of control diet.

Ingredients (%)	Starter: Control 1–21 Days	Grower: Control 21–42 Days
Maize	54.00	59.49
Soy-meal 48%	36.93	31.02
Soybean oil	4.91	5.73
Monocalcium phosphate	1.42	1.30
Calcium carbonate	1.23	1.13
Sodium chloride	0.19	0.21
Sodium bicarbonate	0.27	0.24
DL-methionine	0.30	0.26
L-Lysine HCl	0.23	0.18
Noxyfeed	0.02	0.02
Premix ^1^	0.49	0.44
Calculated content (%)		
Metabolizable energy (Kcal/kg)	3050	3150
Crude protein	22.0	19.5
Ether extract	7.01	7.92
Crude fibre	2.36	2.25
Lysine	1.38	1.18
Methionine + cysteine	0.91	0.87
Threonine	0.81	0.70
Tryptophan	0.21	0.18
Calcium	0.90	0.82
Inorganic phosphorus	0.64	0.59
Sodium	0.16	0.16

^1^ Vitamin-mineral premix provided following nutrients per kg of diet: vitamin A, 13,500 IU; vitamin D3, 4, 800 IU: vitamin E, 67 IU; vitamin B1: 3 mg; vitamin B2, 9 mg; vitamin B6, 4.5 mg; vitamin B12, 16.5 µg; vitamin K3, 3 mg; calcium pantothenate, 16.5 mg; nicotinic acid, 51 mg; folic acid 1.8 mg, biotin: 30 µg; Fe, 54 mg; I, 1.2 mg; Co, 0.6 mg; Cu, 12 mg; Mn, 90 mg; Zn, 66 mg; Se, 0.18 mg; Mo, 1.2 mg.

**Table 2 animals-11-00147-t002:** Collection of biological sample.

Sample	Analysis ^1^
Blood	Hematology
Serum	Response to common vaccines
Plasma	IL-8 Corticosterone DON and DON-3S
Small intestine (Jejunum)	IL-6, IL-1β, IL-10, IFN-^γ^
Liver	DON and DON-3S
Excreta	DON and DON-3S

^1^ IL-8,interleukin 8; DON, deoxynivalenol; DON-3S, deoxynivalenol 3-sulphate; IL-6, interleukin 6; IL-1β interleukin- 1β, IL-10, interleukin 10; IFN-^γ^; interferon gamma.

**Table 3 animals-11-00147-t003:** Mycotoxin analysis of experimental feeds.

Mycotoxin ^1^ (µg/kg)	Control Group		DON Group (5000 µg/kg)		DON Group (15,000 µg/kg)	
	Starter	Grower	Starter	Grower	Starter	Grower
DON	65	73	4760	4650	14,390	15,120
ZEN	<LOD	<LOD	84.4	85.9	242	259
FBs	142	225	257	216	216	275
OTA	0.94	1.59	0.90	1.11	1.21	1.10
AFB1	<LOD	<LOD	<LOD	<LOD	<LOD	<LOD

^1^ DON = deoxynivalenol; ZEN = zearalenone; FBs = fumonisins; OTA = ochratoxin; AFB1 = aflatoxin B1; limit of detection (LOD) of DON, ZEN, FBs, OTA, and AFB1: 50, 1.75, 25, 0.5, 0.3 µg/kg, respectively.

**Table 4 animals-11-00147-t004:** Average concentrations of DON and DON-3S (peak area) in plasma, liver and excreta of broilers fed low DON level (5 mg/kg feed) and high DON level (15 mg/kg feed).

Dietary Treatment/Biological Matrix	DON ^1^	DON-3S ^2^ (×10^6^)
Plasma (ng/mL)		
Control	ND ^3^	ND
DON low level (5 mg/kg)	ND	0.27 ± 0.01 ^b^
DON high level (15 mg/kg)	ND	0.62 ± 0.15 ^a^
SEM	-	0.11
*p*-Value	-	0.01
Liver (ng/g)		
Control	ND	ND
DON low level (5 mg/kg)	ND	ND
DON high level (15 mg/kg)	ND	0.70 ± 0.37
Excreta (ng/g)		
Control	ND	ND
DON low level (5 mg/kg)	22.0	110 ^b^
DON high level (15 mg/kg)	24.1	295 ^a^
SEM	11.9	22.2
*p*-Value	0.81	0.0001

^1^ DON, deoxynivalenol; ^2^ DON-3S, deoxynivalenol 3-sulphate; ^a,b^ within the same column, different superscripts are significantly different (*p* value < 0.05);^3^ ND = not detectable (limit of detection (LOD) = 1.5 ng/mL.

**Table 5 animals-11-00147-t005:** Effects of DON-contaminated feed (5 and 15 mg/kg) on hematological parameters of broiler chickens.

Dietary Treatment ^1^							
Item ^2^	Control	DON (5 mg/kg)	DON (15 mg/kg)	SEM	*p*-Value	Linear	Quadratic
HCT (%)	31.9	30.1	30.2	0.74	0.16	0.15	0.20
HGB (g/dL)	12.1 ^a^	11.1 ^b^	10.1 ^c^	0.29	0.0002	<0.0001	0.54
RBC (×10^6^/µL)	2.3 ^a^	2.3 ^a^	1.9 ^b^	0.38	<0.0001	<0.0001	0.06
MCV (fL)	132 ^b^	133 ^b^	151 ^a^	4.26	0.004	0.001	0.29
MCH (pg)	50.1	49.4	50.5	0.41	0.14	0.22	0.12
MCHC(g/dL)	37.8 ^a^	37.3 ^a^	34.1 ^b^	0.93	0.004	0.001	0.50
Leukocyte (×10^3^/µL)	15.8	18.3	19.2	1.85	0.37	0.20	0.55
Eosinophil (%)	6.00	4.66	7.07	0.93	0.33	0.27	0.32
Basophil (%)	7.84	7.08	6.61	1.32	0.51	0.26	0.76
Lymphocyte (%)	41.2	37.7	37.1	2.81	0.52	0.33	0.55
Monocyte (%)	2.15	2.75	0.92	0.70	0.19	0.17	0.21
Total heterophils (%)	44.5	48.7	48.3	3.17	0.64	0.43	0.61

^1^ DON, deoxynivalenol; SEM, standard error of mean (*n* = 5); ^a,b,c^: means values with different superscripts with the same row differ (*p* ≤ 0.05); ^2^ HCT, hematocrit; RBC, red blood cell; HGB, hemoglobin; MCV, mean corpuscular volume; MCH, mean corpuscular hemoglobin; MCHC, mean corpuscular hemoglobin concentration.

**Table 6 animals-11-00147-t006:** Effects of DON-contaminated feed (5 and 15 mg/kg) on antibody titers against NDV and IBV in broiler chickens.

Dietary Treatment ^1^							
Item ^2^	Control	DON (5 mg/kg)	DON (15 mg/kg)	SEM	*p*-Value	Linear	Quadratic
Titers against NDV (HA)	0.38	0.30	0.25	0.15	0.82	0.56	0.84
Titers against IBV (HA)	2.92	3.20	3.00	0.30	0.81	0.95	0.52
Titers against IBV (ELISA)	874	851	786	120	0.31	0.13	0.76

^1^ DON, deoxynivalenol; SEM, standard error of mean (*n* = 5); ^2^ NDV^,^ Newcastle Disease Virus; HA, haemagglutination inhibition; IBV, infectious bronchitis virus.

**Table 7 animals-11-00147-t007:** Effects of DON-contaminated feed (5 and 15 mg/kg) on the H/L ratio and plasma corticosterone level of broiler chickens.

Dietary Treatment ^1^							
Item ^2^	Control	DON (5 mg/kg)	DON (15 mg/kg)	SEM	*p*-Value	Linear	Quadratic
H/L ratio	1.1	1.3	1.3	0.15	0.47	0.30	0.50
Plasma corticosterone (ng/mL)	2.93 ^a^	2.34 ^b^	2.44 ^b^	0.15	0.03	0.07	0.04

^1^ DON, deoxynivalenol; SEM, standard error of mean (*n* = 5); ^a,b^: values with different superscripts with the same row differ (*p* ≤ 0.05); ^2^ H/L ratio, heterophil to lymphocyte ratio.

## Data Availability

All data sets collected and analyzed during the current study are available from the corresponding author on fair request.

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
