# Peer review of "Effects of Deoxynivalenol-Contaminated Diets on Metabolic and Immunological Parameters in Broiler Chickens"

_animals, 2021, doi:10.3390/ani11010147_

Round 1
Reviewer 1 Report
Theoretically the paper presented for the review meets the criteria for publishing. However I recommend that it could be accepted for the publication after major revisions.
- English - the authors need to improve the English language used in the paper. I suggest in-depth work with native speaker in order to do so. There are uncountable number of typos, sentences that are not properly constructed. Additionally, the use of punctuation needs to be improve, as I have found number of sentences that, from out of nowhere, turned out to finish in the middle.
- The authors cite 63 publications (!!) which in my opinion is way above the standards for a original paper. The outcome of this is that I had a feeling like I was reading a review paper rather that the original one. The discussion section is way too long. Additionally, the authors must clearly state the novelty of their work because I find the results of this investigation as the data that only confirms previous findings. Right now it is hard for me to find any novel data in this paper.
- Table 2 presents results of investigation and it is implemented in the methods section
- Please give more detail on serological examination performed in the study. How was ELISA performed, what IBV and NDV strains were used for HI examination etc.
- Please specify if the Il-8 plasma ELISA kit was specific for chickens. This also concern corticosterone determination
Reviewer 2 Report
Please see the attachment.

Reviewer 3 Report
AN interesting article since DON is the most detected mycotoxin in cereals
Mortality rate in this study was 13, 19, and 13 % for control, DON 5 mg/kg , and DON 15 mg/kg groups accordingly. This rate is quite high, more details are neaded

Round 2
Reviewer 1 Report
Revised manuscript looks very good.
Reviewer 2 Report
The revised manuscript is acceptable.